# Quinone Pool, a Key Target of Plant Flavonoids Inhibiting Gram-Positive Bacteria

**DOI:** 10.3390/molecules28134972

**Published:** 2023-06-24

**Authors:** Li Zhang, Yu Yan, Jianping Zhu, Xuexue Xia, Ganjun Yuan, Shimin Li, Beibei Deng, Xinrong Luo

**Affiliations:** 1Biotechnological Engineering Center for Pharmaceutical Research and Development, Jiangxi Agricultural University, Nanchang 330045, China; 2Laboratory of Natural Medicine and Microbiological Drug, College of Bioscience and Bioengineering, Jiangxi Agricultural University, Nanchang 330045, China

**Keywords:** flavonoids, antimicrobial mechanism, quinone, menaquinone, respiratory chain, bacterium, MIC, *Staphylococcus aureus*, *α*-mangostin

## Abstract

Plant flavonoids have attracted increasing attention as new antimicrobial agents or adjuvants. In our previous work, it was confirmed that the cell membrane is the major site of plant flavonoids acting on the Gram-positive bacteria, which likely involves the inhibition of the respiratory chain. Inspired by the similar structural and antioxidant characters of plant flavonoids to hydro-menaquinone (MKH_2_), we deduced that the quinone pool is probably a key target of plant flavonoids inhibiting Gram-positive bacteria. To verify this, twelve plant flavonoids with six structural subtypes were preliminarily selected, and their minimum inhibitory concentrations (MICs) against Gram-positive bacteria were predicted from the antimicrobial quantitative relationship of plant flavonoids to Gram-positive bacteria. The results showed they have different antimicrobial activities. After their MICs against *Staphylococcus aureus* were determined using the broth microdilution method, nine compounds with MICs ranging from 2 to 4096 μg/mL or more than 1024 μg/mL were eventually selected, and then their MICs against *S. aureus* were determined interfered with different concentrations of menaquinone−4 (MK−4) and the MKs extracted from *S. aureus*. The results showed that the greater the antibacterial activities of plant flavonoids were, the more greatly their antibacterial activities decreased along with the increase in the interfering concentrations of MK−4 (from 2 to 256 μg/mL) and the MK extract (from 4 to 512 μg/mL), while those with the MICs equal to or more than 512 μg/mL decreased a little or remained unchanged. In particular, under the interference of MK−4 (256 μg/mL) and the MK extract (512 μg/mL), the MICs of *α*-mangostin, a compound with the greatest inhibitory activity to *S. aureus* out of these twelve plant flavonoids, increased by 16 times and 8 to 16 times, respectively. Based on the above, it was proposed that the quinone pool is a key target of plant flavonoids inhibiting Gram-positive bacteria, and which likely involves multiple mechanisms including some enzyme and non-enzyme inhibitions.

## 1. Introduction

Antimicrobial resistance (AMR) has brought about a serious threat to public health and economic development, and the COVID-19 pandemic has further accelerated this global problem [1]. Therefore, new antimicrobial agents are being desperately developed [2,3]. Most antibiotics bring about some adverse reactions to the human body during their treatment on bacterial infection, and eventually also become resistant to pathogenic bacteria after a period of use in clinic [4]. However, some plant secondary metabolites not only have antimicrobial activities, but also show a good level of safety for the human body since they exist in all sorts of plant-derived foods and beverages [5,6]. Among them, close attention has been paid to plant flavonoids [7,8,9,10,11]. 

Flavonoids are an important class of secondary metabolites widely distributed in various plants, and approximately 10,000 compounds have been discovered so far. Many of them show different degrees of inhibitory activity to pathogenic bacteria, especially Gram-positive ones, and some of them can also enhance the inhibitory effect of some antimicrobial agents and/or even reverse the AMR [12,13]. Simultaneously, various antibacterial mechanisms were reported for plant flavonoids [7,9,12], which involved the synthesis inhibitions to DNA, proteins and cell envelope, the damage of cell membrane, and so on. Recently, Yuan et al. confirmed that the cell membrane is the major site of plant flavonoids acting on the Gram-positive bacteria, and which includes the damage of phospholipid bilayers and likely involves the inhibition of the respiratory chain, or some others [14,15]. In addition, they pointed out that the antibacterial activities of plant flavonoids to the Gram-positive bacteria are directly related to their lipophilicities, and present nonspecific characterization concluded from the antimicrobial quantitative relationships between the physicochemical parameters and the antimicrobial activities [14,15].

The antimicrobial mechanism of plant flavonoids damaging the phospholipid bilayers of Gram-positive bacteria was confirmed as above, while other mechanisms acting on the cell membrane should be further explored. As Yuan et al. pointed out [14,15], plant flavonoids present a nonspecific antimicrobial mechanism. Therefore, the non-enzyme inhibitions of plant flavonoids to the respiratory chain of Gram-positive bacteria were also our focus, although some probable enzyme mechanisms were likely involved. The compositions of the respiratory chains for different bacteria are varied, while the quinone pool is always a center of electron transfer in the respiratory chain for most bacteria [16,17]. For Gram-positive bacteria, the menaquinone (MK), together with its reducing form as hydro-menaquinone (MKH_2_), is the sole quinone for the electron transfer in the respiratory chain for Gram-positive bacteria [16,17,18]. Inspired by the similar structural and antioxidant characters of plant flavonoids to MKH_2_, we deduced that the quinone pool is a key target of plant flavonoids inhibiting Gram-positive bacteria. To confirm this, here, twelve compounds, with seven structural subtypes and various degrees of inhibitory activities, were preliminarily selected for determining the interference of MK−4 (or the MK extract from *Staphylococcus aureus*) on the inhibitory activities of flavonoids to Gram-positive bacteria.

## 2. Results

### 2.1. Calculated and Tested Minimum Inhibitory Concentrations (MICs) 

As shown on Figure 1, twelve plant flavonoids, with seven structural subtypes including dihydroflavones, flavonols, flavones, isoflavones, chalcones, flavanes and xanthones, were selected for verifying the inference that the quinone pool is a key target of plant flavonoids inhibiting Gram-positive bacteria. Their average MICs (or MIC_90s_) against Gram-positive bacteria such as *S. aureus*, *S. epidermidis* and *Bacillus subtilis* were calculated according to Equation (1) in Section 4.3 [15], and the results (Table 1) showed that they had different degrees of inhibitory activities and can be used for further screening to obtain the required flavonoids with different antibacterial activities. 

Subsequently, the MICs of these plant flavonoids against *S. aureus* ATCC 25923 were determined using the broth microdilution method, and the results are also shown in Table 1. From Table 1, it can be seen that these flavonoids present different degrees of inhibitory activity to *S. aureus* ATCC 25923, with the MIC values ranging from 2 to 4096 (or more than 1024) μg/mL. Considering that selecting plant flavonoids with various structural subtypes and different antimicrobial activities can enhance the scientificity and rationality of the verification experiments, nine plant flavonoids with six subtypes, including *α*-mangostin, sophoraflavanone G, licochalcone A, kurarinone, glabridin, isoliquiritigenin, baicalein, echinatin and quercetin, were selected for further experiments. 

In addition, as observed in Table 1, the results also indicate that the larger the lipophilicities of plant flavonoids, the greater their antimicrobial activities. This confirmed again that the lipophilicity is a key factor of plant flavonoids against Gram-positive bacteria. Furthermore, according to the rules that for the predicted MICs ranging from 1/4× to 4×, the determined ones were acceptable [14,15], and for those less than 1/8× or more than 8×, the determined ones were completely unacceptable, the calculated MIC values of only one plant flavonoid as echinatin was unacceptable, and which once again confirmed the efficiency of Equation (1) for predicting the MICs of plant flavonoids against Gram-positive bacteria. 

### 2.2. Influences of MK−4 on Plant Flavonoids against S. aureus

To verify our hypothesis, MK−4 (menaquinone−4) was selected as a simplified representative for preliminarily exploring the influences of MK−4 on plant flavonoids against *S. aureus*. The results (Figure 2) showed that the antimicrobial activities of five plant flavonoids (*α*-mangostin, sophoraflavanone G, licochalcone A, kurarinone and glabridin), with their MICs ranging from 2 to 16 μg/mL, obviously decreased along with the increase in the interfering concentrations (from 2 to 256 μg/mL) of MK−4 (Figure 2a–e). However, those plant flavonoids with MICs over 512 μg/mL decreased a little or were unable to evaluate (Figure 2f–h) for isoliquiritigenin, baicalein and echinatin, and even remained unchanged for quercetin with the MICs of 4098 μg/mL (Figure 2i). 

In addition, the MIC changes of plant flavonoids against *S. aureus* ATCC 25923 after interfering with the MK−4 concentration of 256 μg/mL are listed in Table 2 for obtaining clearer and more intuitive MIC changes. 

Thus, the above together indicated that the larger the interfering concentrations of MK are, the more remarkably the antibacterial activities of plant flavonoids decrease. Furthermore, it also indicated that the greater the antibacterial activities of plant flavonoids are, the more obvious the interferences of MK on their antibacterial activities are, and the more greatly their antibacterial activities decrease. Simultaneously, menaquinones are the sole quinones in the quinone pools of Gram-positive bacteria, which do not contain ubiquinones. Therefore, it was inferred that plant flavonoids can target the quinone pools of Gram-positive bacteria, especially *S. aureus*.

### 2.3. Influences of MK Extract on Plant Flavonoids against S. aureus

To further confirm the above inference, the MK in the quinone pool of *S. aureus* ATCC 25923 was extracted according to the method in Section 4.6.1 and was marked as an MK extract. Using MK−4 as an internal standard, the HPLC-UV analyses for the MK extract were performed according to the method in [19]. The representative HPLC profile was shown as Figure 3, and its detailed HPLC-UV profile is shown in Appendix A in Supplementary Files. Compared with the UV spectroscopy (Appendix A) of MK−4, for which the retention time is 4.944 min in the HPLC profile (Appendix A), the results (Figure 3 and Appendix A) indicated that there are three menaquinones with the retention times, respectively, at 8.463 (peak 1), 10.535 (peak 2) and 13.318 (peak 3) min in the MK extract. As *S. aureus* mainly contains MK−8 together with a little of MK−7 and −9 in the quinone pool and those MKs have similar physicochemical properties, the main chromatographic peak 2 in the HPLC profile (Figure 3) corresponded to MK−8, and those of peaks 1 and 3 to MK−7 and −9. Therefore, the menaquinones contained in the MK extract were in accordance with those in the quinone pool of *S. aureus* and can be used for further interfering experiments. 

Using this MK extract, the influences of the MK extract on plant flavonoids against *S. aureus* ATCC 25923 were determined. The results (Figure 4) also showed that the antimicrobial activities of five plant flavonoids (*α*-mangostin, sophoraflavanone G, licochalcone A, kurarinone and glabridin) with their MICs ranging from 2 to 16 μg/mL obviously decreased along with the increase in the interfering concentrations of MK−4 from 4 to 512 μg/mL (Figure 4a–e). However, those plant flavonoids with the MICs equal to or more than 1024 μg/mL remained unchanged for isoliquiritigenin and quercetin, respectively, with the MICs of 1024 and 4098 μg/mL (Figure 2f,i), or were unable to evaluate (Figure 2g,h) for baicalein and echinatin. This indicated that the greater the antibacterial activities of plant flavonoids, the more greatly their antibacterial activities of plant flavonoids decrease along with the increase in the interfering concentrations of MK extract. Moreover, for those plant flavonoids with their MICs equal to or more than 1024 μg/mL, their MIC values seemed to remain unchanged along with the increase in the MK extract concentrations from 4 to 512 μg/mL.

In addition, the MIC changes of plant flavonoids against *S. aureus* ATCC 25923 after interfering with the MK−4 concentration of 512 μg/mL are also listed in Table 2 for obtaining clearer and more intuitive change information. From Table 2, can be seen that similar results and rules were presented for the MIC changes of flavonoids inhibiting *S. aureus* whether interfering with the MK extract or MK−4. Namely, the greater the antibacterial activities of plant flavonoids are, the more obvious the interferences of MK extract on their antibacterial activities are, and the more greatly their antibacterial activities decrease. Therefore, it was further confirmed that the quinone pool is a key target of plant flavonoids against *S. aureus* since both of the above results of two interfering experiments were obtained from various structural subtypes with different antimicrobial activities.

## 3. Discussion

In our previous work [14,15], we confirmed that the cell membrane is the main site of plant flavonoids against Gram-positive bacteria, which likely involves respiratory inhibition, etc. Using twelve plant flavonoids including various structural subtypes and different antimicrobial activities, here the interfering experiments of MK−4 and the MK extract from *S. aureus* confirmed that the quinone pool on the respiratory chain is a key target of plant flavonoids against *S. aureus*. 

Similar to *S. aureus*, the menaquinones (MKs) are the sole quinones for electron transfer in the respiratory chain of Gram-positive bacteria [16,17]. Simultaneously, the antimicrobial quantitative relationship between the parameters and the antimicrobial activities, together with many publications [8,13,20], indicated that a certain flavonoid has similar antimicrobial activities to various Gram-positive bacteria. Therefore, it can be inferred that plant flavonoids have a similar mechanism targeting the quinone pools in the respiratory chains of Gram-positive bacteria. For Gram-negative bacteria, there are two quinone MKs and ubiquinones in the quinone pools of their respiratory chains [17], which is in accordance with the fact that plant flavonoids show weak antimicrobial activities to Gram-negative bacteria [8,13,20]. Conversely, this further confirmed the reasonability of our initial inference. 

From Table 1, it can be seen that plant flavonoids with larger lipophilicity, such as *α*-mangostin, sophoraflavanone G, licochalcone A and kurarinone, showed obviously stronger antimicrobial activities than those with smaller lipophilicity, and all compounds with a LogP less than 3.40 scarcely presented antibacterial activity. However, this does not mean that the stronger the lipophilicity, the stronger the antibacterial activity. For example, the lipophilicity (LogP, 4.95) of licochalcone A is weaker than that (LogP, 6.30) of kurarinone, while the antimicrobial activities of the former are stronger than that of the latter. Combined with our previous work [14,15], this further showed that it plays a key role in the antibacterial activities of plant flavonoids to have enough lipophilicity for locating the cell membranes of Gram-positive bacteria and targeting the quinone pool on the respiratory chain. This is also the probable reason that the actual antibacterial activities were inconsistent with those obtained from some experiments at the molecular level [21,22,23,24].

During the interfering experiments of MK−4 and the MK extract from *S. aureus*, some precise MIC data of isoliquiritigenin, baicalein and echinatin were not obtained, and were recorded as more than 1024 μg/mL (Figure 2 and Figure 4) since these flavonoids present too poor a solubility to be effectively determined. However, those of isoliquiritigenin (1024 μg/mL) and quercetin (4096 μg/mL), together with the precise MIC data of other plant flavonoids, already confirmed the MIC change trends of plant flavonoids against Gram-positive bacteria along with the increase in the interfering concentrations of MK−4 or the MK extract. Simultaneously, interfering with the MKs extracted from *S. aureus*, it more powerfully confirmed that the MK pool is a key target of plant flavonoids against Gram-positive bacteria. Furthermore, it is worth exploring whether there are some other lipophilic components, except MKs and membrane phospholipid, with the potency of interference for plant flavonoids against Gram-positive bacteria.

Among these twelve plant flavonoids, *α*-mangostin, a xanthone compound from guttiferaeous plants [25,26], showed the greatest inhibitory activity to *S. aureus*, with an MIC of 2 μg/mL, which is close to the maximum antibacterial activity of plant flavonoids predicted from the antimicrobial quantitative relationship [15]. Therefore, the results from its interfering experiments can best reflect the antibacterial mechanism of plant flavonoids. The results (Table 2) indicate that the MICs of *α*-mangostin to *S. aureus*, respectively, increased by 16 times and 8 to 16 times, under the interference of MK−4 (256 μg/mL) and the MK extract (512 μg/mL). Simultaneously, it was reported that *α*-mangostin can target the bacterial membrane and enhance membrane permeability [27,28,29]. Taken together these powerfully support the idea that the cell membrane is the main site of plant flavonoids against Gram-positive bacteria, involving damage to the phospholipid bilayers and inhibition to the respiratory chain through targeting the quinone pool.

Furthermore, the antimicrobial activities of plant flavonoids decreased along with the interfering concentrations of MK−4 and MK extract. However, this showed that they presented a stepwise decrease (Figure 2 and Figure 3), not a complete dose-dependent *S*-shaped curve. Therefore, the effects of plant flavonoids on the quinone pool of Gram-positive bacteria likely involve multiple mechanisms including enzyme and non-enzyme inhibition. The enzyme mechanisms probably involved the inhibition to some enzymes on the respiratory chain [30], but not on the synthase of MKs since these are the sole quinones in Gram-positive bacteria. The non-enzyme mechanisms probably included the electron transfer, membrane potential and/or reactive oxygen stress (ROS). However, the real ones should be further explored.

## 4. Materials and Methods

### 4.1. Materials, Chemicals and Reagents

Kurarinone (≥98%) was purchased from Wuhan ChemFaces Biochemical Co., Ltd. (Wuhan, China). Sophoraflavanone G (>98%), glabridin (99.8%) and echinatin (98%) were purchased from Shanghai TopScience Co., Ltd. (Shanghai, China). Isoliquirtigenin (98%), formononetin (98%), naringenin (97%), galangin (98%) and baicalin (98%) were purchased from Shanghai Macklin Biochemical Co., Ltd. (Shanghai, China). Quercetin (97%) was purchased from Shanghai Meryer Co., Ltd. (Shanghai, China). Licochalcones A (>98.0%) and *α*-mangostin (>98.0%) were purchased from Chengdu Push Bio-technology Co., Ltd. (Chengdu, China). All the compounds were stored at −20 °C. The stock solutions of the above plant flavonoids were prepared by dissolving them in dimethyl sulfoxide (DMSO) and diluting them with Mueller Hinton broth (MHB) to obtain a concentration of 4096 μg/mL. The stock solution was mixed well and then diluted to the desired concentrations with MHB immediately before use. In another, the DMSO concentrations in all the test systems were kept to less than 5.0%, and all those in the blank controls were 5.0%.

MK−4 was purchased from Sigma-Aldrich (St. Louis, MO, USA). Methanol and petroleum ether used for the MK extract from *S. aureus* were obtained from Xilong Scientific Co., Ltd. (Shantou, China). Casein hydrolysate (Qingdao Hope Bio-Technology Co., Ltd., Qingdao, China), starch soluble (Xilong Scientific Co., Ltd., Shantou, China), beef extract and agar powder (Sangon Biotech (Shanghai) Co., Ltd., Shanghai, China) were used for preparing the media. DMSO was purchased from Sinopharm Chemical Reagent Co., Ltd. (Shanghai, China). Thiazolyl blue tetrazolium bromide was purchased from Sangon Biotech (Shanghai) Co., Ltd. (Shanghai, China), and the 96-well plates were purchased from Shanghai Excell Biological Technology Co., Ltd. (Shanghai, China). All reagents were analytical or biochemical ones. All TopPette Pipettors (2~20 μL and 20~200 μL) were purchased from DLAB Scientific Co., Ltd., Beijing, China.

Mueller Hinton agar (MHA) consisted of casein hydrolysate at 17.5 g/L, starch soluble at 1.5 g/L, beef extract at 3.0 g/L and agar powder at 17.0 g/L dissolved in purified water, with a pH value of 7.40 ± 0.20. MHB was prepared without agar powder according to the same composition and procedure as MHA.

### 4.2. Bacterial Strains and Growth Condition

*S. aureus* ATCC 25923 was purchased from American Type Culture Collection, Manassas, VA, USA, and this organism was stored in Microbank^™^ microbial storages (PRO-LAB diagnostics, Toronto, ON, Canada) at −20 °C. Prior to use, *S. aureus* was cultured onto an MHA plate at 37 °C, and then pure colonies from the plate were inoculated into MHB at 37 °C for 24 h on a rotary shaker (160 rpm). A 1:100 dilution of the overnight culture was made into fresh MHB and then incubated at 37 °C until the exponential phase for the following experiments. MHB was used for the antimicrobial susceptibility tests. 

### 4.3. MIC Calculation

The physicochemical parameters LogP of the tested plant flavonoids were calculated using software ACD/Labs 6.0. Then, the average MIC (or MIC_90_) values of these compounds against Gram-positive bacteria were predicted according to the following Equation (1) [15].
*y* = −0.1285 *x*^6^ + 0.7944 *x*^5^ + 51.785 *x*^4^ − 947.64 *x*^3^ + 6638.7 *x*^2^ – 21,273 *x* + 26,087(1)
where *y* is the average MIC (or MIC_90_) value of a certain flavonoid to Gram-positive bacteria, mainly including *S. aureus*, *S. epidermidis* and *B. subtilis*; *x* is the physicochemical parameter LogP value of this compound.

### 4.4. Antimicrobial Susceptibility Assay

According to the standard procedure described by the Clinical and Laboratory Standards Institute (CLSI) [31], the exponential phase culture was diluted with MHB to achieve an *S. aureus* concentration of approximately 1.0 × 10^6^ CFU/mL, and then the susceptibility of the plant flavonoids against *S. aureus* ATCC 25923 was determined using the broth microdilution method on the 96-well plates (Shanghai Excell Biological Technology Co., Ltd., Shanghai, China) in triplicate [4]. Referring to the calculated MIC values of plant flavonoids, the initial concentration of each compound was set. After the 96-well plates were incubated at 35 °C for 24 h, 20 μL of 3-(4,5-dimethylthiazol-2-yl)-2,5-diphenyltetrazolium bromide (MTT, 4.0 mg/mL) was added into each well, shaken well, and kept for 30 min at an ambient temperature. The minimum inhibitory concentration (MIC), defined as the lowest concentration of compounds that completely inhibited bacterial growth in the micro-wells, was judged from there being no color change when the bacterial growth in blank wells was sufficient [32]. 

### 4.5. Influences of MK−4 on Plant Flavonoids against S. aureus

Using the checkerboard method referring to our previous work [4], the influences of MK−4 on plant flavonoids against *S. aureus* were evaluated from the combined antimicrobial effects of MK−4 and each compound. Briefly, a series of concentrations from 8 to 1024 μg/mL of test compounds (Figure 1) and MK−4, respectively, in the horizontal or vertical direction were prepared with MHB medium in a separate 96-well plate using the twofold dilution method. Next, 50 μL of the test compound or MK−4 with different concentrations was correspondingly added into the designed wells on another plate to obtain different proportions with test compounds (Figure 1) or MK−4 concentrations from 4 to 512 μg/mL, and then 100 μL of bacterial suspension (approximately 1.0 × 10^6^ CFU/mL) was added into each well. In contrast, for compounds **8**, **11** and **12** (Figure 1), the final concentrations of test compounds in corresponding wells ranged from 8 to 1024 μg/mL, and those of compound **6** in Figure 1 ranged from 8 to 4096 μg/mL. 

In addition, column 11 contained a series of concentrations from 2 to 256 μg/mL of MK−4 in MHB with 5 × 10^5^ cfu/mL *S. aureus* isolate, which were used as negative controls. Column 12 contained a series of concentrations from 2 to 256 μg/mL for the test compound (**1**, **2**, **10**, **13** or **14**), from 8 to 1024 μg/mL for the test compound (**8**, **11** or **12**), and from 32 to 4096 μg/mL for compound **6,** which were used as accompanying controls, respectively. According to the same procedure as in Section 4.4, the MICs of each flavonoid against *S. aureus* were determined under the interferences of different MK−4 concentrations.

### 4.6. Influences of MK Extract on Plant Flavonoids against S. aureus

#### 4.6.1. MK Extract from *S. aureus*

Referring to the method reported by Schurig-Briccio et al. [33], the MK was extracted from *S. aureus* ATCC 25923. Briefly, 3000 mL of *S. aureus* cells at the exponential phase was collected via centrifugation at 3000 rpm for 15 min. The pellet was resuspended with 30 mL of purified water, and then the mixture was crushed by a SCIENTZ-IID ultrasound cell breaker (Ningbo Scientz Biotechnology Co., Ltd., Ningbo, China) for 12 min (2 s treatment and 3 s interval). Next, the mixture was centrifuged at 3000 rpm for 15 min, and the pallet was resuspended with 3 mL of water and extracted with 17.5 mL of methanol/petroleum ether (6:4, *v/v*) using a vigorous vortex for 1 min (three times). After being kept for 2 h, the mixture was eddied again for another 1 min, followed by centrifuging at 3000 rpm for 10 min. The upper organic layer was transferred to a 10 mL centrifuge tube and was then evaporated under nitrogen stream to obtain a dried and oily residue (marked as the MK extract). 

#### 4.6.2. HPLC-UV Analyses for the MK Extract

A standard solution (35.0 μg/mL) of MK−4 and a sample solution (128.0 μg/mL) of the MK extract were prepared using methanol/isopropanol (60:40, *v*/*v*). Simultaneously, both the above solutions were mixed in equal volume to obtain a mixed solution of MK−4 plus MK extract, and MK−4 was used as an internal standard. Referring to our previous work [19], the menaquinones contained in the MK extract were analyzed using a HPLC-UV method without methodological validation. Briefly, the HPLC-UV analyses were performed on a Waters e2695 separation system consisting of a model 2998 ultraviolet detector (Milford, MA, USA), and the detection wavelength was set at 247 nm. A Hypersil ODS2 (4.6 mm × 250 mm, 5.0 µm) (Dalian Elite Analytical Instruments Co., Ltd., Dalian, China) was used as the chromatographic column which was kept at room temperature throughout the experiments. Methanol/isopropanol (60:40, *v/v*) was used as the mobile phase, and the flow rate was set at 1.0 mL/min, along with an injection volume of 20 μL. After injection into the HPLC system, the main MKs in the MK extract were identified from the UV spectral characteristics of all chromatographic peaks in the HPLC profile of the mixed solution (Figure 3 and Appendix A), according to our previous publication [19]. In detail, based on the UV spectral characteristics, MK analogs were identified if a chromatographic peak in the HPLC profile of the mixed solution had a similar UV absorption curve to that of the MK−4. 

#### 4.6.3. MICs of Plant Flavonoids with the Interference of the MK Extract

According to the method and procedure in Section 4.5, the MICs of plant flavonoids with the interference of the MK extract were determined using a checkerboard method. In contrast, a series of concentrations from 16 to 2048 μg/mL of the MK extract in the vertical direction were prepared with MHB medium in a separate 96-well plate using the twofold dilution method, and the final concentrations of the MK extract ranged from 4 to 512 μg/mL. In addition, column 11 contained a series of concentrations from 4 to 512 μg/mL of the MK extract in MHB with 5 × 10^5^ cfu/mL *S. aureus* isolate, which were used as negative controls.

## 5. Conclusions

Compared with our previous work showing that plant flavonoids mainly act on the cell membrane of Gram-positive bacteria likely involving the respiratory inhibition, here it was concluded that the quinone pool is a key target of plant flavonoids inhibiting Gram-positive bacteria, which likely involves multiple mechanisms including some enzyme and non-enzyme inhibitions. Moreover, it plays a key role for the antibacterial activities of plant flavonoids in ensuring that they have enough lipophilicity to locate the cell membranes of Gram-positive bacteria and thus target the quinone pool on the respiratory chain.

## Figures and Tables

**Figure 1 molecules-28-04972-f001:**
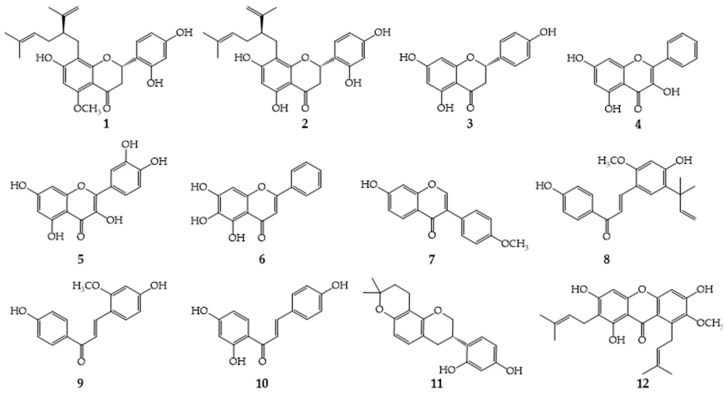
The chemical structures of twelve compounds with seven subtypes of plant flavonoids. **1**, Kurarinone; **2**, Sophoraflavanone G; **3**, Naringenin; **4**, Galangin; **5**, Quercetin; **6**, Baicalein; **7**, Formononetin; **8**, Licochalcone A; **9**, Echinatin; **10**, Isoliquiritigenin; **11**, Glabridin; **12**, *α*-Mangostin.

**Figure 2 molecules-28-04972-f002:**
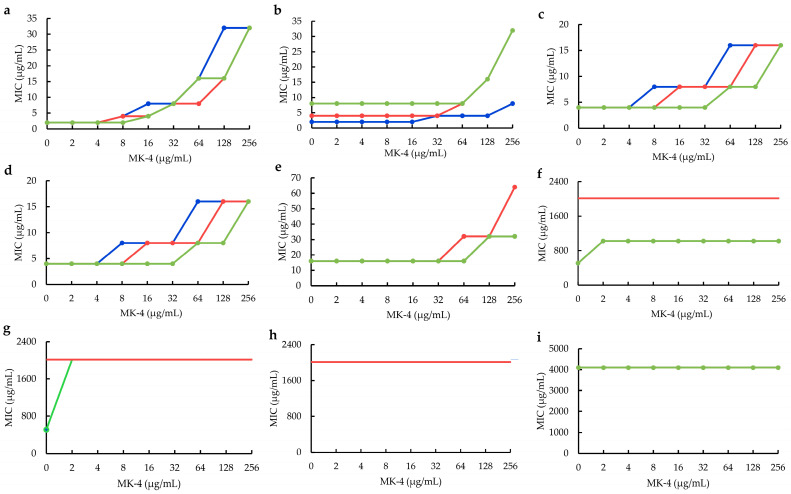
The influences of MK−4 on plant flavonoids against *S. aureus* ATCC 25923. (**a**), *α*-Mangostin; (**b**), Sophoraflavanone G; (**c**), Licochalcone A; (**d**), Kurarinone; (**e**), Glabridin; (**f**), Isoliquiritigenin; (**g**), Baicalein; (**h**), Echinatin; (**i**), Quercetin. Each compound was tested in triple, and tests 1, 2 and 3, respectively, showed the red, blue and green lines in the planes (sometimes the red, blue and/or green lines overlapped and only showed as the red, blue or green lines). Those lines without data dots in (**f**–**h**) indicated that the MICs were more than 1024 μg/mL.

**Figure 3 molecules-28-04972-f003:**
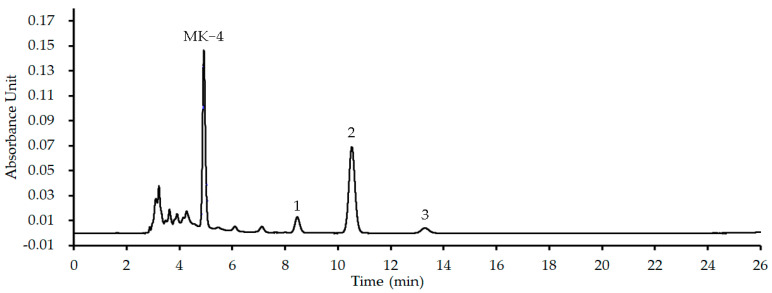
The HPLC profile of representative MK extract from *S. aureus* ATCC 25923. MK−4 was used as an internal standard, and peaks 1, 2 and 3 corresponded to MK−7, −8 and −9, respectively.

**Figure 4 molecules-28-04972-f004:**
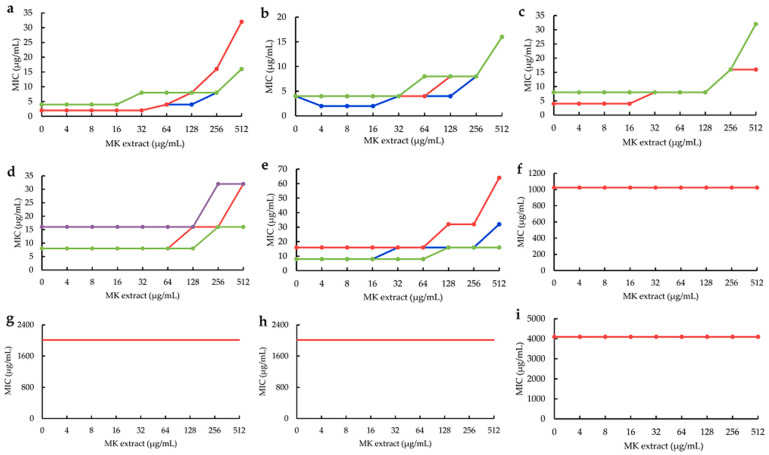
The influences of MK extract on plant flavonoids against *S. aureus* ATCC 25923. (**a**), *α*-Mangostin; (**b**), Sophoraflavanone G; (**c**), Licochalcone A; (**d**), Kurarinone; (**e**), Glabridin; (**f**), Isoliquiritigenin; (**g**), Baicalein; (**h**), Echinatin; (**i**), Quercetin. Each compound was tested in triple, and tests 1, 2 and 3, respectively, showed the red, blue and green lines in the planes (sometimes the red, blue and/or green lines overlapped and only showed as the red, blue or green lines). Those lines without data dot in (**f**–**h**) indicated the MICs were more than 1024 μg/mL.

**Table 1 molecules-28-04972-t001:** Minimum inhibitory concentrations (MICs) of twelve plant flavonoids against Gram-positive bacteria.

Compounds	LogP	Calculated MICs ^a^		Tested MICs (μg/mL) ^b^	Compounds	LogP	Calculated MICs		Tested MICs (μg/mL)
		μmol/L	μg/mL				μmol/L	μg/mL	
Kurarinone	6.30	28.92	12.67	8	Formononetin	3.15	549.61	147.35	˃1024
Sophoraflavanone G	6.52	16.37	6.95	2~4	Licochalcone A	4.95	74.44	25.17	4
Naringenin	3.19	509.64	138.67	512	Echinatin	3.23	472.20	127.54	˃1024
Galangin	2.83	974.47	263.15	>1024	Isoliquiritigenin	3.40	338.78	86.76	512~1024
Quercetin	2.07	3063.61	925.21	4096	Glabridin	4.39	74.74	24.38	8~16
Baicalein	3.31	404.47	109.23	512~˃1024	α-Mangostin	6.70	8.17	3.35	2

^a^: Considered as the average MIC (or MIC_90_) values of plant flavonoids against Gram-positive bacteria such as *S. aureus*, *S. epidermidis* and *B. subtilis* were calculated from Equation (1); ^b^: the MIC values of plant flavonoids against *S. aureus* ATCC 25923 were determined in triplicate.

**Table 2 molecules-28-04972-t002:** The MIC changes of plant flavonoids against *S. aureus* after interfering with maximum test concentrations of MK−4 and MK extract ^a^.

Compounds	MIC_Alone_ (μg/mL)	MIC Change (Times) ^b^		Compounds	MIC_Alone_ (μg/mL)	MIC Change (Times)	
		MK−4	MK Extract			MK−4	MK Extract
α-Mangostin	2	16	8~16	Isoliquiritigenin	512~˃1024	2/- ^c^	1
Sophoraflavanone G	2~4	4~8	4~8	Baicalein	512~˃1024	-	-
Licochalcone A	4	4	4~8	Echinatin	˃1024	-	-
Kurarinone	8	2~4	2~4	Quercetin	4096	1	1
Glabridin	8~16	2~4	2~4				

^a^: The test microorganism is *S. aureus* ATCC 25923; ^b^: the interfering concentrations of MK−4 and MK extract were 256 and 512 μg/mL, respectively; ^c^: - indicated that the increased times were uncertain since no definite MIC value was obtained.

## Data Availability

Not applicable.

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
