# Peer review of "Quinone Pool, a Key Target of Plant Flavonoids Inhibiting Gram-Positive Bacteria"

_molecules, 2023, doi:10.3390/molecules28134972_

Round 1

Reviewer 1 Report

The abstract adequately presents the research conducted.

The introduction justifies the direction of the research, the latest literature sources are analyzed, a suitable research plan ensures the achievement of the goal, and the inhibitory concentrations of flavonoids are determined.

Appropriate test cultures are selected, research results are compared with new literature sources.The abstract adequately presents the research conducted.

Author Response

Dear Reviewer,

My co-authors and I are very grateful to you for your careful review and good comments. Based on the original manuscript, we have carefully checked throughout the manuscript again, and have pleasure to submit the revised version, together with the responses to your comments, for your consideration.

Many thanks for your kind attention!

Yours sincerely,

Ganjun Yuan

Here are our answers to your comments.

1. The abstract adequately presents the research conducted.

Response 1: Thank you for your careful review and good comments!

2. The introduction justifies the direction of the research, the latest literature sources are analyzed, a suitable research plan ensures the achievement of the goal, and the inhibitory concentrations of flavonoids are determined.

Response 2: Thank you for your careful review and good comments!

3. Appropriate test cultures are selected, and research results are compared with new literature sources. The abstract adequately presents the research conducted.

Response 3: Thank you for your careful review and good comments!

Other revision

We had updated Figures 2 and 4 by adding the results from the third interfering experiments which further confirmed the results in the original manuscript. Moreover, we had carefully performed extensive revision again throughout the manuscript including all references, the spelling, formatting, syntax, linguistic edit and expressions, for your consideration.

Reviewer 2 Report

Li Zhang et al. present an interesting in vitro study focusing on antimicrobial activity of plant flavonoids including gram-positive bacteria and the site of their action.

The text is easily readable and understandable, the methods are described in detail. My comments are mainly of technical character as follows:

-          In the abstract, abbreviations should be provided with their full names first, then in brackets. Thus, the abstract would be more comprehensive for non-specialists in the field

-          Please, correct the names of bacteria S. aureus, S. epidermidis and Bacillus subtilis to Italic format throughout the MS, e.g.: lines 106, 110, 113, 130, 131, 135, 149, 152, 154, 164, 166, 178, 195, 216, 219, 224 etc.

-          Line 141, 201 – please, correct to remained

-           What I understand is Mueller-Hinton agar and broth media were prepared in the lab, why commercially available media were not purchased?

-          Line 303 “and then pure colonies from the plate were inoculated into MHA at 303 37°C for 24 h on a rotary shaker (160 rpm)”; into MHA or MHB?

-          Line 352 – 3,000 mL or 3 mL?

-          Line 353 – correct to pellet

Author Response

Dear Reviewer,

My co-authors and I are very grateful to you for your careful review, good comments, kind reminder and valuable suggestions. We have amended the manuscript according to the issues raised by you, and have pleasure to submit the revised version, together with the responses to your comments, for your consideration.

Many thanks for your kind attention!

Yours sincerely,

Ganjun Yuan

Here are our answers to your comments.

Li Zhang et al. present an interesting in vitro study focusing on antimicrobial activity of plant flavonoids including gram-positive bacteria and the site of their action. The text is easily readable and understandable, the methods are described in detail.

Response: Thank you for your careful review and good comments!

1. In the abstract, abbreviations should be provided with their full names first, then in brackets. Thus, the abstract would be more comprehensive for non-specialists in the field

Response 1: Thank you for your careful review, kind reminder and valuable suggestions! We had already checked them and provided their full names together with their abbreviations in brackets when they first appeared. Thank you very much!

2. Please, correct the names of bacteria S. aureus, S. epidermidis and Bacillus subtilis to Italic format throughout the MS, e.g.: lines 106, 110, 113, 130, 131, 135, 149, 152, 154, 164, 166, 178, 195, 216, 219, 224 etc.

Response 2: Thank you for your careful review and kind reminder! We had already revised them as Italic format throughout the manuscript.

3. Line 141, 201 – please, correct to remained

Response 3: Thank you for your careful review and kind reminder! We had already revised them.

4. What I understand is Mueller-Hinton agar and broth media were prepared in the lab, why commercially available media were not purchased?

Response 4: Thank you for your careful review and kind reminder! You are right! Mueller-Hinton agar (MHA) and broth (MHB) media can be commercially available. Since we previously found that the agar plates cannot solidify well when using a commercial MHA medium, sometimes we have prepared them in the lab. Correspondingly, MHB media are also prepared in the same experiments.

5. Line 303 “and then pure colonies from the plate were inoculated into MHA at 303 37°C for 24 h on a rotary shaker (160 rpm)”; into MHA or MHB?

Response 5: Thank you for your careful review and kind reminder! You are right! The term MHA should be revised as MHB, and we had already revised it. Thank you very much!

6. Line 352 – 3,000 mL or 3 mL?

Response 6: Thank you for your careful review and kind reminder! We had checked it, and the volume of culture is 3,000 mL.

7. Line 353 – correct to pellet

Response 7: Thank you for your careful review and kind reminder! We had already revised it.

Other revision

We had updated Figures 2 and 4 by adding the results from the third interfering experiments which further confirmed the results in the original manuscript. Moreover, we had carefully performed extensive revision again throughout the manuscript including all references, the spelling, formatting, syntax, linguistic edit and expressions, for your consideration.

Reviewer 3 Report

The manuscript presents the results of a study aimed at proving that plant flavonoids are capable of inhibiting the growth of gram-positive bacteria by affecting their respiratory chain, specifically the quinone pool. The results presented support the hypothesis.

The manuscript is interesting, but needs definite improvement.

Major remarks

Figure 1 with commentary should be at the beginning of section 2.1, not at the end of the introduction.

Discussion is vague, partly a repetition of the information in the Introduction, this acct is unnecessary. This point needs to be reworked, specific information about compounds active against bacteria should be given, for example, in Abstract the compound a-mangostin is mentioned, highlighting its activity, while Discussion does not even mention it, why?

Conclusions is another repetition of information already known from the Introduction and Discussion. This section is supposed to contain a specific summary of the results, not vague information.

Minor remarks

All names of bacterial strains should be written in italics.

Line 110 - ATCC25923 should be written together or separately?

Line 279 - what is meant by the term "certain volume"

Line 141 and 201 - instead of remianed should be remained

Line 168 - instead of extract should be extract

Line 179 - should be „can be used”

Author Response

Dear Reviewer,

My co-authors and I are very grateful to you for your careful review, good comments, kind reminder and valuable suggestions. We have amended the manuscript according to the issues raised by you, and have pleasure to submit the revised version, together with all the responses to your points, for your consideration.

Many thanks for your kind attention!

Yours sincerely,

Ganjun Yuan

Here are our answers to your comments.

The manuscript presents the results of a study aimed at proving that plant flavonoids are capable of inhibiting the growth of gram-positive bacteria by affecting their respiratory chain, specifically the quinone pool. The results presented support the hypothesis.

The manuscript is interesting, but needs definite improvement.

Response: Thank you for your careful review, good comments, kind reminder and valuable suggestions.

Major remarks

1. Figure 1 with commentary should be at the beginning of section 2.1, not at the end of the introduction.

Response 1: Thank you for your careful review and valuable suggestions! You are right! It is more scientific and reasonable for laying Figure 1 at the beginning of section 2.1. According to your suggestion, we had already revised it.

2. Discussion is vague, partly a repetition of the information in the Introduction, this acct is unnecessary. This point needs to be reworked, specific information about compounds active against bacteria should be given, for example, in Abstract the compound a-mangostin is mentioned, highlighting its activity, while Discussion does not even mention it, why?

Response 2: Thank you for your careful review and valuable suggestions! We had deleted most repetition information in section “Introduction” and/or improved our expressions. According to your suggestions, we had inserted two paragraphs in section “Discussion” as follows Lines 266-277 and 290-301 in the revised manuscript. Correspondingly, we had added nine references in section “References”. Thank you for your help to improve our work! 

Lines 266-277: “From Table 1, it was found that plant flavonoids with larger lipophilicity, such as α-mangostin, sophoraflavanone G, licochalcone A and kurarinone, showed obviously stronger antimicrobial activities than those with smaller lipophilicity, and all compounds with the LogP less than 3.40 scarcely presented antibacterial activity. However, it does not mean that the stronger the lipophilicity, the stronger the antibacterial activity. For example, the lipophilicity (LogP, 4.95) of licochalcone A is weaker than that (LogP, 6.30) of kurarinone, while the antimicrobial activities of former are stronger than that of latter. Combined with our previous work [14,15], this further showed that it plays a key role for the antibacterial activities of plant flavonoids to have enough lipophilicity for locating the cell membranes of gram-positive bacteria and targeting the quinone pool on the respiratory chain. This is also the probable reason that the actual antibacterial activities were inconsistent with those obtained from some experiments at molecular level [21-24].”

Lines 290-301: “Among these twelve plant flavonoids, α-mangostin, a xanthone compound from guttiferaeous plants [25,26], showed greatest inhibitory activity to S. aureus, with the MIC of 2 μg/mL which is close to the maximum antibacterial activity of plant flavonoids predicted from the antimicrobial quantitative relationship [15]. Thereby, the results from its interfered experiments can best reflect the antibacterial mechanism of plant flavonoids. The results (Table 2) indicated that the MICs of α-mangostin to S. aureus respectively increased by 16 times and 8 to16 times, under the interference of MK-4 (256 μg/mL) and the MK extract (512 μg/mL). Simultaneously, it was reported that α-mangostin can target bacterial membrane and enhance membrane permeability [27-29]. These together powerfully supported that the cell membrane is the main site of plant flavonoids against gram-positive bacteria, involving the damage to the phospholipid bilayers and the inhibition to the respiratory chain through targeting the quinone pool.”

3. Conclusions is another repetition of information already known from the Introduction and Discussion. This section is supposed to contain a specific summary of the results, not vague information.

Response 3: Thank you for your careful review, kindly reminder and valuable suggestions! According to your suggestions, we had deleted most repetition information in section “Introduction” and “Discussion”, and improve this section as follows for your consideration:

“Go deep into our previous work that plant flavonoids mainly act on the cell membrane of gram-positive bacteria likely involving the respiratory inhibition, here it was concluded that the quinone pool is a key target of plant flavonoids inhibiting gram-positive bacteria, and which likely involves multiple mechanisms including some enzyme and non-enzyme inhibitions. Moreover, it plays a key role for the antibacterial activities of plant flavonoids that they have enough lipophilicity to locate the cell membranes of gram-positive bacteria and thus target the quinone pool on the respiratory chain.”

Thank you for your help to improve our work!

Minor remarks

4. All names of bacterial strains should be written in italics.

Response 4: Thank you for your careful review and valuable suggestions! We had already revised them as Italic format throughout the manuscript.

5. Line 110 - ATCC25923 should be written together or separately?

Response 5: Thank you for your careful review and kind reminder! We had already revised it as ATCC 25923.

6. Line 279 - what is meant by the term "certain volume"

Response 6: Thank you for your careful review and kind reminder! We had deleted the term “a certain volume of” for clearer expression.

Comments on the Quality of English Language

7. Line 141 and 201 - instead of remianed should be remained

Response 7: Thank you for your careful review and kind reminder! We had already revised them.

8. Line 168 - instead of extarct should be extract

Response 8: Thank you for your careful review and kind reminder! We had already revised it.

9. Line 179 - should be „can be used”

Response 9: Thank you for your careful review and kind reminder! We had already revised it.

Other revision

We had updated Figures 2 and 4 by adding the results from the third interfering experiments which further confirmed the results in the original manuscript. Moreover, we had carefully performed extensive revision again throughout the manuscript including all references, the spelling, formatting, syntax, linguistic edit and expressions, for your consideration.

Round 2

Reviewer 3 Report

The authors have answered my questions and made suggested corrections. Thank you, I have no further comments.